# Increased Immunoglobulin and Proteoglycan Synthesis in Resected Hippocampal Tissue Predicts Post-Surgical Seizure Recurrence in Human Temporal Lobe Epilepsy

**DOI:** 10.3390/pathophysiology32020015

**Published:** 2025-04-14

**Authors:** Michael F. Hammer, Martin E. Weinand

**Affiliations:** 1BIO5 Institute, University of Arizona, Tucson, AZ 85721, USA; mfh@arizona.edu; 2Department of Neurology, University of Arizona College of Medicine, Tucson, AZ 85724, USA; 3Department of Neurosurgery, University of Arizona College of Medicine, Tucson, AZ 85724, USA

**Keywords:** temporal lobe epilepsy, surgery outcome, autoantibodies, neuroinflammation, transcriptome, temporal lobectomy

## Abstract

**Background/Objectives:** For patients with medically refractory temporal lobe epilepsy (TLE), surgery is an effective strategy. However, post-operative seizure recurrence occurs in 20–30% of patients, and it remains challenging to predict outcomes solely based on clinical variables. Here, we ask to what extent differences in gene expression in epileptic tissue can predict the outcome after resective epilepsy surgery. **Methods:** We performed RNAseq on hippocampal tissue resected from eight patients who underwent anterior temporal lobectomy with amygalohippocampectomy (ATL/AH), half of whom became seizure free (SF) or non-seizure free (NSF). **Results:** Bioinformatic analyses revealed 1548 differentially expressed genes and statistical enrichment analyses identified a distinct set of pathways in NSF and SF cohorts that were associated with neuroinflammation, neurotransmission, synaptic plasticity, and extracellular matrix (ECM) reorganization. Resected tissue exhibiting strong pro-inflammatory processes are associated with better post-surgery seizure outcomes than patients exhibiting cellular signaling processes related to ECM reorganization, autoantibody production, and neural circuit formation. **Conclusions:** The results suggest that post-operative targeting of both inhibitory aspects of the ECM remodeling and the autoimmune/inflammatory components may be helpful in promoting repair and preventing the recurrence of seizures.

## 1. Introduction

Temporal lobe epilepsy (TLE) is the most prevalent and medically intractable form of adult partial epilepsy [1]. TLE produces recurrent seizures originating from the amygdala and hippocampus complex and parahippocampal region [2,3]. About one-third of patients with TLE become drug resistant [4,5]. For patients with refractory seizures in spite of therapeutic anti-seizure medications (ASMs), treatment with epilepsy surgery may offer a potential cure. TLE surgery includes ablation or resection of epileptogenic temporal lobe tissue, including stereotactic laser amygdalohippocampotomy (SLAH) or anterior temporal lobectomy with amygdalohippocampectomy (ATL/AH), respectively [6]. Patients may be rendered seizure free in about ~60% of cases with ablative surgery (i.e., SLAH) and in approximately 80% of cases with ATL/AH [7,8].

The extent of resection, pathology type, epilepsy duration, and localization pattern may be important determinants of post-surgical seizure control [9,10,11,12,13]. However, it has remained challenging to predict post-operative seizure freedom solely based on these features. Other factors may extend beyond traditional clinical variables, such as those involving cellular or molecular processes that lower seizure threshold [14]. An important unanswered question is to what extent differences in gene expression in epileptic hippocampal tissue can predict the outcome after resective epilepsy surgery. There is still limited direct evidence on specific cellular or signaling pathway differences in resected brain tissue that correlate with seizure freedom versus seizure recurrence after temporal lobe epilepsy surgery.

Transcriptome studies have identified altered gene expression patterns in tissues resected from patients with TLE [6,14,15,16,17]. These observations suggest that recurrences after epilepsy surgery may be influenced partly by differences in neuroinflammatory and/or neuronal healing/remodeling pathways. Here, we perform RNAseq on hippocampal tissue from eight patients, four of whom remained seizure free (SF) and four of whom experienced seizure recurrence (NSF) after surgery. Analyses of the altered genome-wide patterns of transcript abundance reveal several commonalities as well as stark differences between these two cohorts. Our results suggest that resected tissue exhibiting strong pro-inflammatory processes are associated with better post-surgery seizure outcomes than patients exhibiting cellular signaling processes related to extracellular matrix (ECM) reorganization, autoantibody production, and neural circuit formation.

## 2. Materials and Methods

### 2.1. Patient Samples

The University of Arizona Institutional Review Board approved all research consent and the protocol for this study. Eight patients with medically intractable complex partial epilepsy underwent right-sided anterior temporal lobectomy with hippocampectomy (ATL/AH), from which hippocampal tissue was obtained. ATL/AH included resection of at least 5.5 cm of right-lateral temporal cortex. The en bloc hippocampal resection extended posteriorly to at least the level of the cerebral peduncle. The hippocampus was preserved for gene expression analysis in the manner described previously [18].

### 2.2. RNASeq and Statistical Analyses

Statistical results are conveyed as the mean ± Standard Deviation. We used a “perturbation signature” approach to identify genome-wide differences in transcript abundance between patients that were not seizure free (NSF) and those that were seizure free (SF) after surgery. We compared RNA-sequencing findings in our non-seizure-free cohort (NSF) with those of the seizure-free (SF) cohort to filter out common alterations due to having epilepsy per se. RNAseq and differential expression analyses were performed as previously described [15,19]. A stranded mRNA-Seq kit, with assessed average fragment size, was used to construct the libraries. Rapid-Run SBS 2 x 100 bp chemistry was used to perform sequencing on the HiSeq2500 (Illumina, San Diego, CA, USA). Five healthy subjects provided post mortem control human hippocampal tissue RNA-seq data [20]. Htseq-count version 0.6.1 was used to obtain gene expression counts [21]. We normalized gene expression counts with the calcNormFactors function of the exactTest function in edgeR. During each sequencing run, we obtained about 20 million high quality sequencing reads, with >90% aligned with the reference genome. We evaluated all reads, producing for differential expression analysis a total of 63,677 final transcripts. Differential expression gene (DEG) analysis was performed on NSF versus SF groups using four biological replicates per group, which has been shown in power analyses to be sufficient to yield a true positive rate greater than 80% under the conditions used here [22]. Only genes that were differentially expressed between NSF and SF at a level of ≥2.0 log2FC were included in analyses (i.e., n = 1548 DEGs).

We identified significantly deactivated or activated biological pathways, with involvement of putative upstream transcriptional regulators, with analysis of significant differentially expressed genes (DEGs), defined by false discovery rate (FDR) ≤ 0.01 for NSF and SF versus controls, FDR  ≤  0.05 for NSF versus SF using Ingenuity^®^ Pathway Analysis (IPA) (Qiagen, Hilden, Germany), and *p*-value ≤ 0.05. This RNAseq bioinformatic analysis determined the most significantly enriched biological pathways based on differentially expressed genes (DEGs). Using a set of log2-transformed counts calculated by the rlog function in DESeq2, we conducted principal component analysis (PCA) with DESeq2’s plotPCA function [23]. Based on the read counts of the 500 genes with greatest expression variance, PCA plots were created.

To predict whether pathways were activated or deactivated relative to baseline, we also included comparisons of NSF and SF samples versus a set of controls derived from RNA-seq data of five healthy post mortem human hippocampal tissues [20]. To ensure that our results were not overly biased by altered gene expression due solely to post mortem processes, we reported results for pathways that were shown to be significantly altered only in the NSF versus SF comparison (i.e., the intersection of all three comparisons).

### 2.3. Pathway Literature Searches and Feature Selection

To aid evaluation of IPA-identified canonical pathways, we conducted a literature search using “Claude” AI assistant (Anthropic, San Francisco, CA, USA). We submitted the following query: “what does activation/deactivation of ‘canonical pathway name’ have to do with brain injury?” We identified several pathway “*effects*” (e.g., blood–brain barrier, neuroinflammation) in the SF and NSF subjects. We assigned a score of 1 or 2 to each canonical pathway *effect* based on ‘detrimental’ and ‘beneficial’ aspects, respectively, in association with either deactivated or activated states. For instance, with the pathway *effect* “neuroinflammation”, either a ‘1’ or ‘2’ was designated when pro- or anti-inflammatory cytokines were produced early and/or late during the injury process, respectively. The score of ^1^/_2_ was designated when there was a beneficial *effect* in one context but a detrimental *effect* in another context. A given pathway was designated as “0” when the *effect* was not involved in that pathway. We fundamentally assumed that samples of tissue were obtained in chronic epileptogenic stages for the assignment of whether an *effect* was detrimental or beneficial. A final list of references cited for each search was constructed and evaluated for result verification in pathways determined to be the most relevant.

## 3. Results

### 3.1. Cohort Characteristics

The ages of the six males and two females spanned 16 to 38 years (mean = 29.0 ± 7.0). The NSF and SF cohorts each had three males and one female, with mean ages of 29.0 ± 5.4 and 29.0 ± 9.3 years, respectively (Table 1). The etiology of seizures included drug overdose, stroke, and febrile seizures, with five of unspecified origin. An equal number of patients in the NSF and SF cohorts had signs of hippocampal sclerosis and hippocampal atrophy. All patients recorded seizure activity for at least 8 years prior to surgery (mean seizure duration = 17.9 ± 11.2 years), with NSF and SF averaging 20.7 ± 7.2 and 15.8 ± 14.2 years, respectively. Pre-surgical seizure frequency data were collected for each patient, which ranged from 0.33 to 10 seizures per month (mean = 5.3 ± 3.1), with NSF and SF averaging 6.0 ± 2.3 and 4.6 ± 4.0 seizures/month, respectively. Patients were taking an average of 3.8 ± 2.3 anti-seizure medications (ASMs), with NSF and SF averaging 4.3 ± 2.5 and 3.3 ± 2.2 ASMs, respectively. Patients were followed clinically for 34.5 ± 27.9 months post-operatively, while SF patients were followed for a minimum of 12 months (mean = 23.5 ± 10.2 months). There were no statistically significant differences in these features between the NSF and SF cohorts.

### 3.2. Principal Component Analysis

Figure 1 shows the results of a principal component analysis that was conducted using the 500 transcripts with the greatest variance to determine the largest source of variation in the data for the NSF versus SF comparison. The first two principal components explain ~63% of the total variation in gene expression. Three of the four NSF subjects cluster tightly on the upper left portion of the plot with one subject as an outlier in the upper right side. The four SF subjects span the central portions of the plot (Figure 1A).

We then performed a similar analysis after including RNA-seq data from five “controls” that derive from five adult autopsy samples from hippocampal tissue [20]. In this case, a total of 65% of the variance was explained in the first two dimensions, with the five controls positioned on the right side of the plot (Figure 1B). A similar pattern was obtained for NSF and SF, with three of the four NSF subjects clustering in the upper left and a single outlier at the bottom left. The four SF subjects clustered between these two points. These results suggest that seizure freedom is one of the main determinants of variance within the expression data.

### 3.3. Differential Expression Analysis

In comparing gene expression differences between NSF and SF, 1548 genes were significantly differentially expressed (*p*-value ≤ 0.05, FDR ≤ 0.05), 952 with elevated transcript abundance in NSF (≥2-fold, range 2.0–457) and 632 with lower transcript abundance (≥2-fold, range 2.0–98.5). To infer the number of genes that were upregulated and downregulated, we performed comparisons of transcript abundance in NSF versus controls and SF versus controls. A total of 1082 transcripts were upregulated and 484 downregulated in NSF (FDR-adjusted *p*-value ≤ 0.05) (Figure 2A). The SF versus control analysis yielded a larger number of differentially expressed genes (DEGs): 1385 upregulated and 716 downregulated. There were 506 upregulated and 205 downregulated DEGs shared between NSF and SF, respectively.

### 3.4. Canonical Pathways Altered in NSF and SF Cohorts

We investigated the results of pathway enrichment procedures, limiting reporting to significant results shared among comparisons of NSF vs. SF (FDR ≤ 0.05) and NSF or SF versus controls (FDR ≤ 0.01) and z-scores with absolute values ≥ 2.0 (Figure 3). For the NSF cohort, pathway enrichment procedures identified 37 canonical pathways, all of which were predicted to be activated (Figure 2B). A total of 35 significantly altered canonical pathways were identified in the SF cohort, 17 of which were predicted to be deactivated and 18 to be activated. Shared canonical pathways between NSF and SF included two predicted to be activated in NSF and deactivated in SF (Adrenergic Receptor Signaling and Neurexins and Neuroligins) and seven predicted to be activated in both (S100 Family Signaling, Interleukin 17A (IL-17A) Signaling in Fibroblasts, Macrophage Alternative Activation Signaling, Class A/1 Rhodopsin-like receptors, Interleukin-10 signaling, cell surface interactions at the vascular wall and Inducible nitric oxide synthase (iNOS) Signaling) (Figure 2B and Figure 3).

The top five pathways predicted to be solely activated in NSF include Binding and Uptake of Ligands by Scavenger Receptors, G-Protein Coupled Receptor Signaling, Potassium Channels, cAMP response element-binding protein (CREB) Signaling in Neurons, and G alpha (q) signaling events. For SF, the top unique activated pathways include Pathogen Induced Cytokine Storm Signaling, Atherosclerosis Signaling, Triggering receptor expressed on myeloid cells-1 (TREM1) Signaling, Toll-like Receptor Signaling, and High mobility group box 1 (HMGB1) Signaling (Figure 4). Atherosclerosis is the main cause of ischemic stroke and cardiovascular disease and is considered an inflammatory disease—providing a pathway link with epileptogenesis [14,24]. The top five deactivated pathways for SF were Glutaminergic Receptor Signaling, Synaptogenesis Signaling, Synaptic Long-Term Potentiation, Dopamine–dopamine and cAMP-regulated phosphoprotein (DARPP32) Feedback in cAMP Signaling, and Netrin Signaling.

### 3.5. Hierarchical Pathway Categories

Figure 4 shows a bubble chart representing a subset of the 63 pathways (n = 36 “selected” pathways) that most strongly distinguish NSF and SF in terms of enrichment *p*-value and activation score. The SF cohort is characterized by a nearly equal representation of activated and deactivated pathways, many of which function in immune and neuronal systems, while NSF shows a pattern dominated by activation of pathways associated with adaptive immune, neuronal, and ECM functionality. These higher categories can be visualized in Figure 5A, where we classify all 63 significantly enriched pathways into one of six higher-level functional categories: immune, metabolism, development, neuronal, signaling, and extracellular matrix (ECM). Nearly equal percentages of NSF-activated pathways fall under immune and neuronal categories, followed by signaling and ECM (Figure 5B). In contrast, the majority of SF deactivated pathways are in the neuronal category (Figure 5C), and nearly all of the SF activated pathways are immune related (Figure 5D).

### 3.6. Predicted Upstream Regulators

To identify potential drivers of the differential expression patterns observed within each dataset, we used the upstream regulator function in IPA. In general, the top predicted activated molecules had higher z-scores for the SF compared with the NSF cohort (top five mean = 8.6 versus 6.7, respectively). Lipopolysaccharide (LPS) was the top predicted activator for both NSF and SF (z-score = 7.8 and 10.7, respectively). However, Cyclic-AMP response element binding protein 1 (CREB1) was the next top predicted upstream activator for the NSF cohort (z-score = 6.9), while the immune-related molecules, Interleukin-1 beta (IL1B) and Tumor necrosis factor (TNF), were the next top predicted activators for the SF cohort (z-score = 8.8 and 8.2, respectively).

### 3.7. Pathway Effects

In Figure 3, for the 52 canonical pathways unique to NSF or SF cohorts, we conducted literature searches for cellular processes commonly described in brain injuries and additional neurological diseases. We tallied the number of times each of the 13 resulting effects were involved in all 52 pathways. Hallmarks of neurological disease or prominent roles in disease pathogenesis or brain injury response were evident for many of the identified pathway effects [25,26]. In 37 of the 52 (71.2%) pathways, ‘neuroinflammation’ was identified as the most common pathway (i.e., the sum of INF in first column of Figure 3C,D).

This was followed by ‘cell survival/apoptosis’ (SVL, 67.3%), ‘synaptic plasticity’ (SNP, 65.4%), ‘repair/recovery’ (REP, 63.5%), ‘cognition’ (CGN, 40.4%), ‘excitotoxicity’ (ETX, 38.5%), ‘blood-brain barrier’ (BBB, 36.5%), ‘secondary injury cascade’ (SIC, 32.7%), ‘oxidative stress’ (OXS, 32.7%), ‘axon guidance/neurite outgrowth’ (AXG, 28.8%), ‘neurotransmission’ (NTM, 21.2%), ‘glial/fibrotic scar’ (GFS, 19.2%), and ‘extracellular matrix’ (ECM, 13.5%) (Figure 3).

NSF versus SF effect frequencies differ; five pathway effects are higher in NSF compared with SF, INF (84.6% vs. 57.7%), SVL (80.8% vs. 53.8%), AXG (34.6% vs. 23.1%), GFS (30.8% vs. 7.7%), and ECM (26.9% vs. 0.0%), respectively, while three are more frequent in SF, REP (76.9% vs. 50.0%), CGN (61.5% vs. 19.2%), and BBB (50.0% vs. 23.1%), respectively (Figure 6A). The mean number of effects associated with a pathway is 2.7 ± 2.5. For the combined cohort, the mean number of beneficial effects (2.1 ± 2.4) is lower than that for detrimental effects (3.2 ± 2.5) (*t*-test *p*-value = 0.010) (Figure 3). This trend is more extreme for the SF cohort, which has a much lower mean number of beneficial (0.9 ± 1.9) versus detrimental effects (4.3 ± 2.0) (*t*-test *p*-value ≤ 1 × 10^−5^). The NSF cohort shows the opposite trend with a larger mean number of beneficial effects (3.5 ± 2.2) compared with detrimental effects (1.9 ± 2.3) (*t*-test *p*-value = 6.3 × 10^−3^) (Figure 3). This shift toward increased detrimental effects in the SF cohort is clearest for SVL, CGN, OXS, and NTM (Figure 3).

### 3.8. Divergent Immune System- and Neuronal System-Related Pathway Effects

As reported above, NSF and SF cohorts differ in the activation status of immune- and neuronal-related pathways (Figure 5). A striking difference between NSF and SF hippocampal tissue is the over-expression of nearly three dozen immunoglobulin genes in NSF. Of the 40 immunoglobulin DEGs in both the NSF and SF cohorts, 34 were found to be over-expressed in NSF (Table 2), with only 2 expressed at lower levels. The genes uniquely upregulated in NSF include several that can form autoantibodies. The top hit, when performing an overrepresentation test for these 34 DEGs in the Reactome knowledgebase, was ‘complement cascade’. Our analysis reveals distinct effects of deactivation/activation of these pathways, whereby NSF has a mixed beneficial to detrimental profile for immune-related pathways and a predominantly beneficial profile for neuronal-related pathways. In contrast, SF has a detrimental profile for both immune- and neuronal-related pathway alterations (Figure 6B). Further subclassification of immune and neuronal functions indicates SF pathways characterized by detrimental activation of pro-inflammatory cytokines and inflammatory immune cells, with no evidence of adaptive immune/antibody-mediated processes as seen in NSF (Figure 6D). NSF has several activated pathways that function in neurotransmission and neuronal excitability with chiefly beneficial effects (Figure 6C), while SF has several deactivated pathways that function in the formation of neural circuits (axon guidance, neurite outgrowth), synaptic plasticity, and neurotransmission—all with mainly detrimental effects in the context of injured brain tissue (Figure 6D).

## 4. Discussion

While clinical features have shown utility in predicting post-operative outcomes [11,27,28], the process of evaluating which temporal lobe epilepsy (TLE) patients are the best surgical candidates based solely on clinical/imaging data is complex [29]. For example, models to predict surgical outcomes have <75% discrimination [16,28]. The incorporation of molecular data to improve clinical decision-making has shown promise [17,30,31,32], including recent genome-wide approaches. Hershberger et al. [16] found that the upregulation of genes related to immune response and inflammation was associated with a higher risk of seizure recurrence and that altered expression of genes involved in synaptic transmission and neuronal plasticity was associated with a lower risk of seizure recurrence. Similarly, Louis et al. [17] found that upregulation of genes related to neuroinflammation, glial cell activation, and oxidative stress was associated with seizure recurrence and downregulation of genes involved in Gamma-aminobutyric acid (GABA) ergic signaling and synaptic plasticity was associated with seizure freedom. Focusing more on risks of late seizure recurrence, Jehi et al. [14] found associations related to neuronal plasticity, synaptic transmission, and immune response. These findings underscore the known complexity of epilepsy pathophysiology and molecular mechanisms that may contribute to post-surgical seizure outcomes. The main unanswered question is how localized alterations of these pathways (i.e., in the resected tissue itself) predict post-surgical outcomes (i.e., more widespread and/or persistent effects).

Our transcriptome analyses of resected hippocampal tissue identified several cellular signaling pathways that distinguish non-seizure-free (NSF) from seizure-free (SF) patients. In the following sections, we attempt to infer how these alterations predict or influence post-operative seizure freedom versus seizure recurrence. We do not favor an explanation based solely on pre-surgical clinical factors because evaluations of our patients were performed at a single neurosurgical center, and NSF and SF cohorts did not vary significantly in clinical features that have been shown to be predictors of surgical outcomes [9,11,12,25] (Table 1).

### 4.1. Shared Pathways Indicative of Common Processes in Epileptogenesis

We used a two-prong strategy to identify molecular and cellular signaling processes that may explain post-surgical seizure outcomes. By comparing genome-wide changes in transcript abundance in NSF versus SF, we filtered out many common alterations that underly the epileptogenic process per se. Similar rates of anti-seizure medication (ASM) usage, epilepsy duration, and seizure frequencies help to homogenize the impact of these variables on the genetic makeup of the tissue [14]. The drawback of using internal cohort comparisons is that they cannot determine whether genes are upregulated or downregulated relative to baseline expression levels expected under physiological conditions. Therefore, we also compared each TLE cohort with a set of autopsy controls without epilepsy [20]. The results reported in Figure 2, Figure 3, Figure 4, Figure 5 and Figure 6 represent only those genes and pathways that were statistically significant in both analyses.

### 4.2. Pathways Distinguishing NSF from SF

Deactivated and activated pathways in hippocampal tissue from the NSF cohort primarily fell into four higher-level categories: extracellular matrix (ECM), immune, neuronal, and signaling systems (Figure 5A,B). This pattern contrasts with that of the SF cohort, which primarily exhibited the deactivation of neuronal pathways and the activation of immune pathways (Figure 5C,D). Moreover, the ratio of beneficial to detrimental pathway effects differed significantly for the higher-level categories shared between the NSF and SF cohorts (Figure 6B). Neuronal system pathway effects were generally inferred to be beneficial in the NSF and detrimental in the SF cohort (Figure 3 and Figure 6B). While immune-related pathways were of mixed effect in NSF, they were exclusively associated with detrimental effects in the SF cohort (Figure 6B). In the next section, we discuss pathways and pathway effects that uniquely characterize the NSF and SF cohorts to identify potential factors underlying seizure freedom or recurrence.

### 4.3. Increased Expression of Immunoglobulins in NSF

A striking difference between NSF and SF hippocampal tissue is the over-expression of nearly three dozen immunoglobulin genes in NSF. Of the 40 immunoglobulin differentially expressed genes (DEGs) in both NSF and SF cohorts, 34 were found to be over-expressed in NSF (Table 2), with only two expressed at lower levels. The genes uniquely upregulated in NSF include several that can form autoantibodies. The top hit when performing an overrepresentation test for these 34 DEGs in the Reactome knowledgebase was ‘complement cascade’. The complement system is a critical part of the immune response that enhances the ability of antibodies and phagocytic cells to clear damaged cells and promote inflammation. Antibodies that participate in autoimmunity can interact with the complement cascade in several significant ways, contributing to both physiological immune function and the pathology of autoimmune diseases. In fact, our results bear some resemblance to a recent study that found increased inflammatory response characterized by the activation of the complement system in drug-resistant TLE patients who experienced seizure recurrence after hippocampectomy [26].

How might increased immunoglobulin expression, particularly in the context of autoimmunity and persistent inflammation, contribute to the risk of post-surgery seizure recurrence in TLE and other neurological disorders? Autoantibodies targeting neuronal proteins (e.g., anti-N-methyl-D-aspartic acid (NMDA) receptor; anti-Gamma-aminobutyric acid (GABA) receptor) can have widespread effects on neural circuit function and plasticity, contributing to the development and persistence of seizures even after surgical resection of the epileptic focus [33]. In spinal cord injuries, B cells producing pathogenic antibodies have been shown to impair recovery [34], and a spinal cord injury can trigger systemic autoimmunity, characterized by chronic B lymphocyte activation and autoantibody synthesis [35,36]. The high-affinity IgG receptor, Fc gamma receptor I (FcγRI), has also been implicated in modulating neuropathic pain after a peripheral nerve injury [37], suggesting a role for immunoglobulins in the pathogenesis of neurological disorders.

The risk of developing autoimmune epilepsy after TLE surgery may be elevated in patients with an increased expression of immunoglobulins in the resected hippocampal tissue [38,39]. Such increased expression may suggest the presence of pre-existing autoimmune activity, even if it has not yet manifested as clinical autoimmune epilepsy [40]. Thus, it is plausible that autoimmune epilepsy could potentially be triggered by the resection of hippocampal tissue in TLE patients after surgery and affect other brain regions [41]. In other cases, the exposure of neuronal antigens during surgery may lead to the production of autoantibodies that cross-react with other proteins in the brain due to molecular mimicry [42].

### 4.4. The Role of Increased Activation of Chondroitin and Dermatan Synthesis in NSF

Chondroitin sulfate proteoglycans (CSPGs) are key components of the ECM in the central nervous system (CNS). Increased activation of chondroitin and dermatan synthesis, particularly after a CNS injury or in neurological disorders like epilepsy, can lead to the formation of a dense, inhibitory CSPG-rich matrix [43,44]. This inhibitory matrix can limit synaptic plasticity, axonal growth, and regeneration, thus hindering repair processes and functional recovery [45,46,47]. Interestingly, the enzymatic digestion of CSPGs using chondroitinase ABC (ChABC) has shown promise in promoting functional recovery and reducing pathology in spinal cord injury models [45,48]. In the context of temporal lobe epilepsy, increased chondroitin 6-sulfation has been implicated in the formation of aberrant neural circuits and the persistence of seizures [49]. The upregulation of CSPGs can also contribute to the formation of a glial scar, which acts as a physical and chemical barrier to regeneration [48,50].

### 4.5. What Factors Explain the Association of Increased Pro-Inflammatory Markers and Post-Surgery Seizure Freedom?

In contrast to our NSF patients, patients in the SF cohort had increased levels of neuroinflammatory markers involved in innate immunity, the production of pro-inflammatory cytokines, the regulation of inflammation, and the recruitment of immune cells to the site of injury. The activation of these pathways is consistent with findings in other TLE studies, especially in patients with hippocampal sclerosis (HS) [51,52]. Interestingly, cells of adaptive immunity such as T and B cells or natural killer (NK) cells were not detected in human epileptic tissue [52], which differentiated brain inflammation in TLE from inflammation in Rasmussen’s encephalitis, where cells of adaptive immunity are strongly represented in the lesional tissue [53]. The association of activated neuroinflammatory processes in patients with better post-surgical outcomes appears to be contrary to the aforementioned transcriptome studies [14,16,17]. Figure 6 shows a significant presence of detrimental inflammatory pathways in SF compared with NSF. Current surgical practices appear to be resecting inflammatory hippocampal pathophysiology in SF patients and “ineffectively” resecting adaptive/antibody-mediated pathophysiology in NSF patients. Continued research is needed to establish whether the increased expression of neuroinflammatory markers and signaling processes in resected tissue could be an indicator of a more localized pathology that is amenable to surgical treatment (as opposed to increased CSPGs and immunoglobulin levels, which may have a more extensive and complex pathology that is less likely to be resolved by surgery alone).

### 4.6. Implications for Pre- and Post-Surgery Surveillance

Future work is needed to determine whether these results have implications for pre- and post-surgery surveillance such as (1) identifying patients with a higher likelihood of post-surgery seizure freedom and (2) monitoring patients after surgery and treatment when increased expression of CSPGs and immunoglobulins is identified in resected tissue. Incorporating circulating inflammatory cytokine (along with CSPG and immunoglobulin expression data, if possible) into pre-surgical evaluation protocols might help guide patient counseling, surgical decision-making, and post-operative management strategies. Indeed, gene expression profiles are known to differ in pre-surgical peripheral blood samples taken from NSF and SF patients, and several candidate biomarkers have been identified [54]. For example, BGN codes for biglycan, which is a structural component of the ECM that also acts as a danger signal that stimulates multifunctional pro-inflammatory signaling, linking the innate to the adaptive immune response [55]. Other genes with potential prognostic value include several with the ECM and/or inflammation-related roles (e.g., Bridging Integrator 3 (BIN3); Matrix Metallopeptidase 8 (MMP8); Interferon Alpha Inducible Protein 27 (IFI27); Interleukin 22 Receptor Subunit Alpha 1 (IL22RA1); Radical S-Adenosyl Methionine Domain Containing 2 (RSAD2); Platelet Factor 4 Variant 1 (PF4V1)) as well as with roles in synaptic plasticity and repair (e.g., Proteolipid Protein 1 (PLP1); Glial Fibrillary Acidic Protein (GFAP); Nectin Cell Adhesion Molecule 2 (PVRL2); Cytoplasmic Polyadenylation Element Binding Protein 4 (CPEB4); and MAM Domain Containing Glycosylphosphatidylinositol Anchor 1 (MDGA1)) [54].

Validating these results on a biological level may also inform researchers on patients at risk for seizure recurrence and other neurological deficits after surgery. In addition to standard ASMs, some patients may benefit from immunomodulatory therapies, such as intravenous immunoglobulin (IVIG) or plasmapheresis, which have been used to treat autoimmune epilepsy and other neurological disorders associated with pathogenic autoantibodies [41,56]. It may also be of benefit to consider continued monitoring of circulating cytokines with a neuroinflammatory profile during long-term follow-up as an additional tool that may be used as an indication of favorable outcomes after temporal lobe surgery [51].

## 5. Conclusions and Limitations

In summary, this study revealed a distinct set of immunological and neurological processes in hippocampal tissue resected from patients who became seizure free or who experienced recurrent seizures following temporal lobectomy and amygdalohippocampectomy. In both the NSF and SF cohorts, there is ample evidence of alterations involving the regulation of neuroinflammation. However, the immune response in the NSF cohort is mainly geared to the clearance of cellular debris and potentially harmful molecules through phagocytosis and antibody-dependent cellular cytotoxicity, while the response in the SF cohort is aimed at the production of pro-inflammatory cytokines and chemokines, which can exacerbate neuroinflammation and secondary damage in pathologic brain tissue. There is a similar division in the neuronal response, with NSF pathways primarily involved in the modulation of neuronal excitability and neurotransmission, the activation of which may have a neuromodulatory effect. On the other hand, many neuronal pathways were deactivated in the SF cohort, which may have the effect of impairing synaptic function, reducing neurotransmitter release, and disrupting the formation of neural circuits. Finally, this study identified pathways that involve chondroitin and dermatan synthesis that were activated in the NSF cohort, which may have a more widespread inhibitory effect on neuronal plasticity.

An important open question for future work is whether the increased activation of these pathways and the concomitant upregulation of immunoglobulin expression can work in concert to inhibit repair processes and contribute to the persistence of seizures after surgery. Targeting both the inhibitory CSPG matrix and the autoimmune/inflammatory components may be necessary to effectively promote repair and prevent the recurrence of seizures and other neurological deficits [57]. While uncovering many pathological processes that may be prognostic for surgery outcomes in patients with TLE, we point out two limitations of this study, including a relatively small cohort size and the lack of suitable control tissue. While we prioritized the NSF versus SF comparisons, we also relied on post mortem hippocampal tissue to predict whether pathways were activated or deactivated relative to the baseline at physiological conditions. It is also important to note that future studies are needed to validate these findings at the biological level and to test the potential role of autoimmunity in the post-surgical non-seizure-free condition. Additional functional studies, including with animal models of temporal lobe epilepsy, are needed to replicate and confirm the relationships between canonical pathways and beneficial and detrimental effects in non-seizure-free and seizure-free conditions. Finally, the discovery of biomarkers in samples that are easily accessed from patients in presurgical examinations (e.g., peripheral blood, including leukocytes) will greatly facilitate the translation of this study. Leukocyte gene expression has been shown to predict post-operative seizure freedom for a variety of genes and biological pathways involved in temporal lobe epilepsy pathogenesis [54]. To further investigate the prognostic value of immunoglobulins for seizure recurrence, an analysis of systemic leukocyte gene expression for immunoglobulins (Table 2) might serve as a pre-surgical validation step for correlation with post-operative seizure outcomes.

## 6. Patents

Dr. Weinand is the author of US Non-Provisional Patent Application 18/754,715 (UA23-142, ARIZ 23.12 NP), “Methods for the prognosis and treatment of temporal lobe epilepsy”, submitted 26 June 2024.

## Figures and Tables

**Figure 1 pathophysiology-32-00015-f001:**
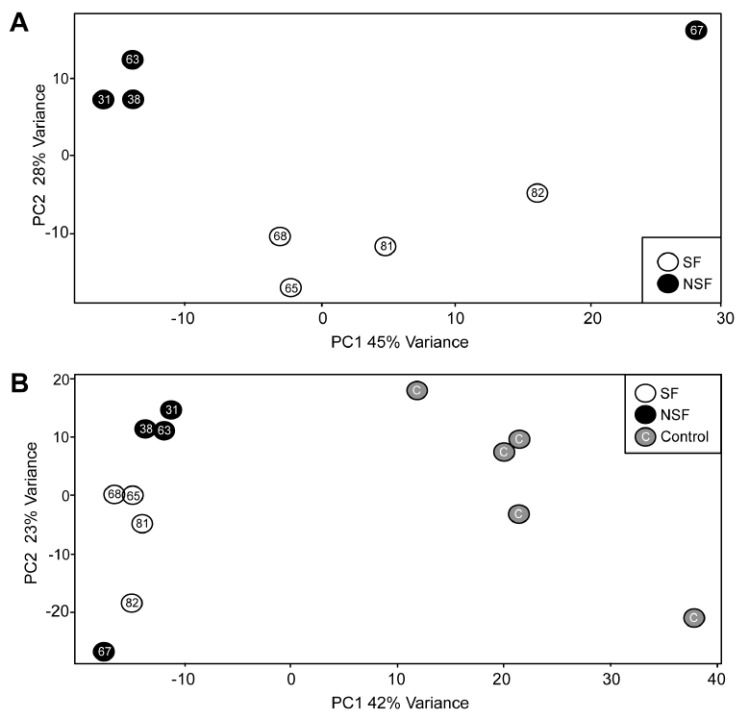
Principal component analysis (PCA). (**A**) The PCA based on the 500 transcripts with the greatest variance in expression. Black-filled circles represent NSF individuals, and open circles represent SF individuals. The first two principal components explain ~73% of the total variation in transcript abundance. (**B**) The PCA including 5 post mortem control samples based on the 500 transcripts with the greatest variance in expression. Gray-filled circles represent controls. The first two principal components explain 65% of the variance in transcript abundance.

**Figure 2 pathophysiology-32-00015-f002:**
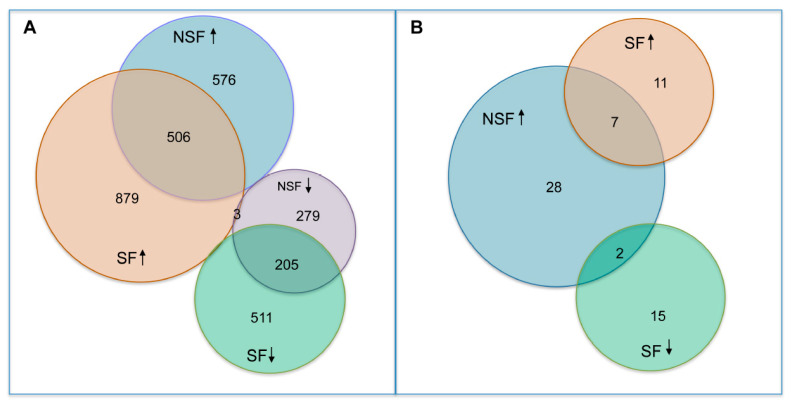
Number of differentially expressed genes and pathways identified by RNAseq and pathway enrichment analysis. (**A**) Venn diagram showing number of differentially expressed genes (DEGs) identified in NSF and SF (**B**) Venn diagram showing number of significantly enriched canonical pathways identified by IPA in NSF and SF.

**Figure 3 pathophysiology-32-00015-f003:**
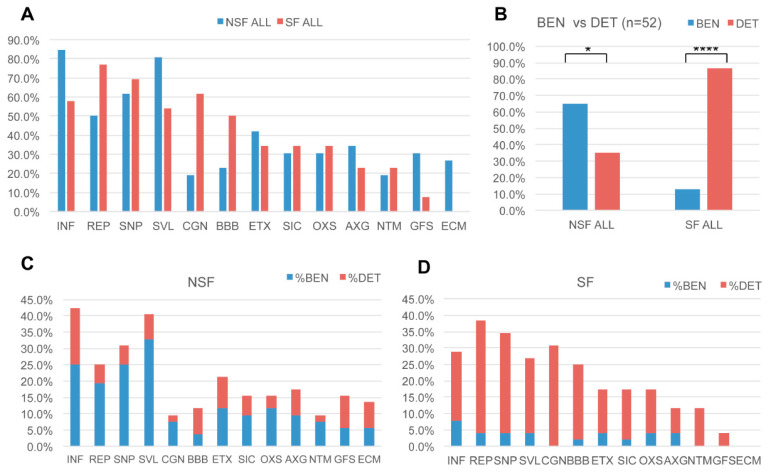
Pathway effects in NSF and SF cohorts. (**A**) Bar chart showing frequency of 13 common pathway effects representing a range of injury processes associated with 54 canonical pathways that distinguish NSF and SF. Neuroinflammation (INF); repair (REP); synaptic plasticity (SNP); survival (SVL); cognition (CGN); the blood–brain barrier (BBB); excitotoxicity (ETX); secondary injury cascade (SIC); oxidative stress (OXS); axonal guidance (AXG); neurotransmission (NTM); glial activation/fibrosis (GFS); and the extracellular matrix (ECM). (**B**) Mean frequency of beneficial (BEN) and detrimental (DET) effects for NSF and SF pathways. *t*-test: * *p* = 0.05; **** *p* < 0.0001. (**C**) The ratio of beneficial and detrimental effects associated with NSF pathways and (**D**) SF pathways.

**Figure 4 pathophysiology-32-00015-f004:**
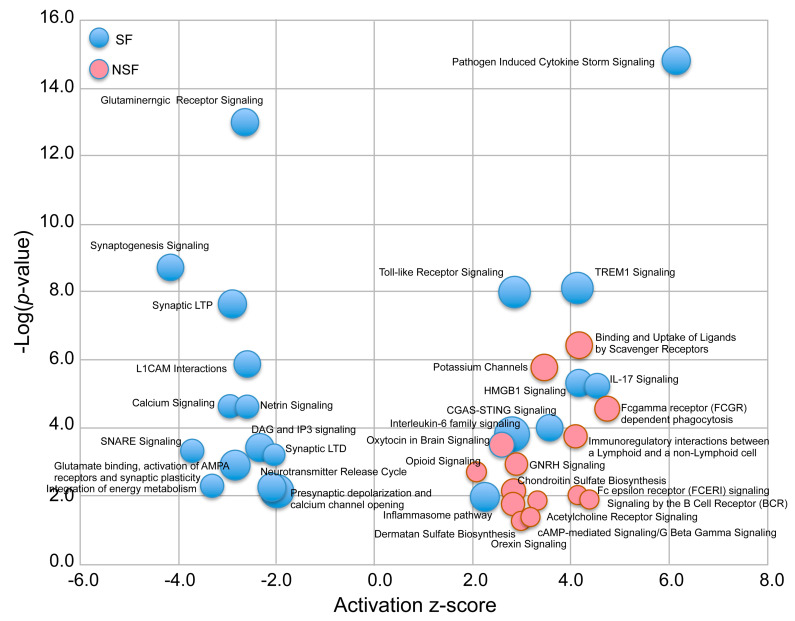
A bubble chart showing the 36 most highly significantly enriched pathways (−log (*p*-value) ≥ 1.3 and activation z-score ≤ −2.0 or ≥2.0) common to the three comparisons: NSF versus SF, NSF versus controls, and SF versus controls. The radius of the circles represents the ratio of differentially expressed genes (DEGs) to total number of genes in each pathway.

**Figure 5 pathophysiology-32-00015-f005:**
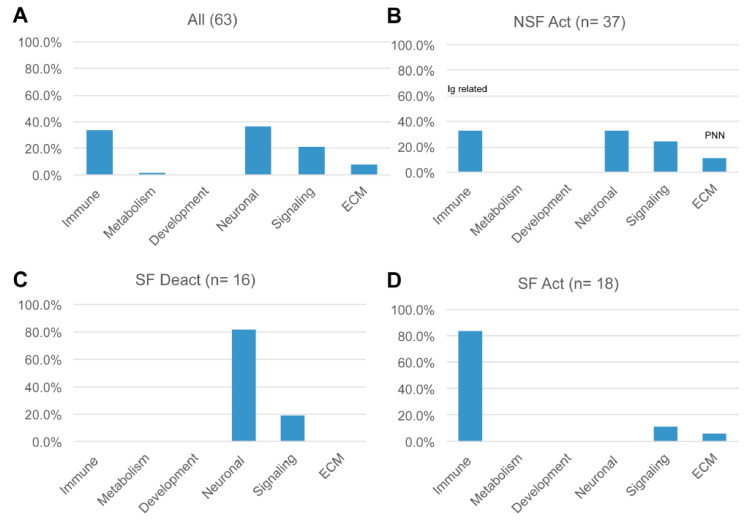
The number of canonical pathways in each of the six higher-level functional categories. (**A**) The percentage of deactivated NSF pathways, (**B**) percentage of activated NSF pathways, (**C**) percentage of deactivated SF pathways, and (**D**) percentage of activated SF pathways. Percentage values are the number of pathways in each category relative to the total number of deactivated or activated pathways for the NSF or SF cohorts, which are listed at the top of each bar chart. Deactivated pathways (Deact), activated pathways (Act).

**Figure 6 pathophysiology-32-00015-f006:**
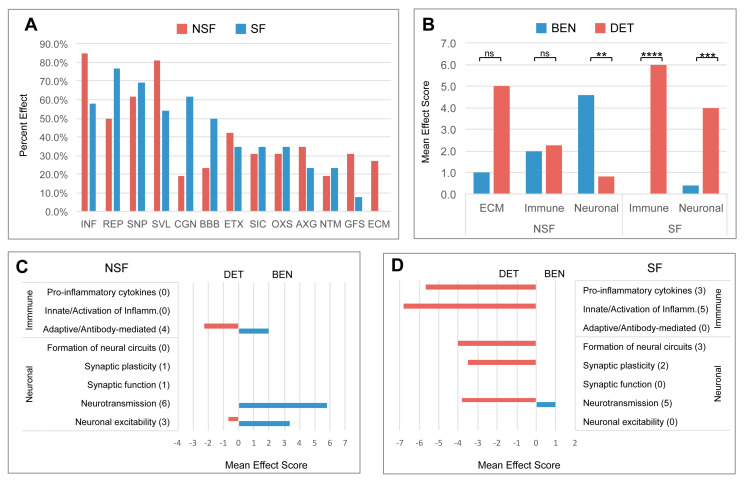
Beneficial and detrimental pathway effects in NSF and SF cohorts. (**A**) Bar chart showing 13 common pathway effects representing a range of injury processes associated with 52 distinguishing canonical pathways that distinguish NSF and SF. Neuroinflammation (INF); repair (REP); synaptic plasticity (SNP); survival (SVL); cognition (CGN); blood–brain barrier (BBB); excitotoxicity (ETX); secondary injury cascade (SIC); oxidative stress (OXS); axonal guidance (AXG); neurotransmission (NTM); gliosis/fibrotic scar (GFS); and extracellular matrix (ECM). (**B**) Mean frequency of beneficial and detrimental effects associated with higher-level-ECM-, neuronal-, and immune-related categories. *t*-test: NSF ECM *p* = 0.156 (ns); NSF immune *p* = 0.903 (ns); NSF neuronal *p* = 0.002 (**); SF immune *p* < 0.0001 (****); SF neuronal *p* = 0.001 (***). (**C**) The mean frequency of beneficial and detrimental effects associated with NSF pathways within neuronal- and immune-related subcategories (the number of individual pathways in each subcategory is shown in parenthesis). (**D**) The mean frequency of beneficial and detrimental effects associated with SF pathways within neuronal- and immune-related subcategories (the number of individual pathways in each subcategory is shown in parenthesis); ns = not significant.

**Table 1 pathophysiology-32-00015-t001:** Clinical characteristics of patients with refractory temporal lobe epilepsy (TLE).

Subject #	Surgery Outcome	Sex	Age (yr)	Duration (yr)	Etiology	Onset Age (yr)	Seizures/Mo	H.S.	#ASMs
1	NSF	M	32	17	Unk	15	4	No (HA)	8
2	NSF	M	32	29	Unk	3	4	no	3
3	NSF	M	21	Unk	Unk	Unk	8	yes	3
4	NSF	F	31	16	Unk	15	8	yes	3
5	SF	F	32	8	CVA	24	0.33	No (HA)	1
6	SF	M	16	10	Unk	6	4	yes	6
7	SF	M	30	8	OD	22	4	yes	4
8	SF	M	38	37	Feb	1	10	No (HA)	2

Surgery Outcome: Not seizure free (NSF), seizure free (SF); sex: male (M), female (F); duration of seizures prior to surgery (Duration); etiology of epilepsy (Etiology): stroke (CVA); drug overdose (OD*); febrile illness (Feb); unknown (Unk); age of onset of epilepsy (Onset age); frequency of seizures per month prior to surgery (Seizures/mo); hippocampal sclerosis (H.S.), hippocampal atrophy (HA); number of antiseizure medications (#ASMs); years (yr). *OD = drug overdose with 3,4-methylenedioxymethamphetamine (MDMA). # = number.

**Table 2 pathophysiology-32-00015-t002:** Differentially expressed immunoglobulin genes in NSF and SF patients ^1^.

Category	Gene Name	NSF vs. SF	NSF vs. Control	SF vs. Control	Role in Autoantibody Function
IgH	IGHV5-51	159.0	527.6		Antigen specificity, can target self
	IGHV4-39	131.8	103.6		Antigen specificity, can target self
	IGHG3	127.6	156.9		IgG component; autoimmunity
	IGHV1-18	118.1	202.6		Antigen specificity, can target self
	IGHG1	107.9	449.0	4.2	IgG component; autoimmunity
	IGHV2-5	96.4	216.9		Antigen specificity, can target self
	IGHV1-69	47.8	2117.3	44.6	Antigen specificity, can target self
	IGHG4	38.3	206.0		IgG component; autoimmunity
	IGHGP	31.1	45.7		Pseudogene, no autoantibody role
	IGHV3-7	20.0	102.1		Antigen specificity, can target self
	IGHM	6.6	4.7		IgMcomponent; form autoantibodies
	IGHV1-46	5.5	64.5		Antigen specificity, can target self
	IGHA1		8.8		IgAcomponent; form autoantibodies
	IGHV1-67		46.4		Antigen specificity, can target self
	IGHV3-33		32.4		Antigen specificity, can target self
IgL	IGLC3	195.9	58.3		Lambda light chains, autoantibodies
	IGLV2-14	139.6	52.0		Antigen specificity, can target self
	IGLV1-44	96.4	65.2		Antigen specificity, can target self
	IGLC2	39.4	195.9		Lambda light chains, autoantibodies
	IGLV1-40	28.4	55.7		Antigen specificity, can target self
	IGLV4-69		42.7		Antigen specificity, can target self
IgK	IGKV1-39		443.2		Antigen specificity, can target self
	IGK1-5	456.9	169.4		Antigen specificity, can target self
	IGK1-9		23.7		Antigen specificity, can target self
	IGKV1D-33		32.4		Antigen specificity, can target self
	IGKV1D-39	17.7	269.8		Antigen specificity, can target self
	IGKV3-11	65.1	13.2		Antigen specificity, can target self
	IGKV3-15		12.6		Antigen specificity, can target self
	IGKV3-20	251.4	570.8		Antigen specificity, can target self
	IGKV3D-15		131.6		Antigen specificity, can target self
	IGKV4-1	37.3	7.3		Antigen specificity, can target self
Others	IGSF9B	2.5	2.8		Not directly in autoantibodies
	ISLR		2.2		Not directly in autoantibodies
	ISLR2		2.2		Not directly in autoantibodies

^1^ Numbers represent statistically significant fold-changes after false discover rate adjustment (*p* ≤ 0.05).

## Data Availability

The original data presented in the study are openly available and have been uploaded to the University of Arizona Figshare account at https://doi.org/10.25422/azu.data.28782032 (accessed on 9 April 2025).

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
