# Peer review of "Increased Immunoglobulin and Proteoglycan Synthesis in Resected Hippocampal Tissue Predicts Post-Surgical Seizure Recurrence in Human Temporal Lobe Epilepsy"

_pathophysiology, 2025, doi:10.3390/pathophysiology32020015_

Round 1

Reviewer 1 Report

Comments and Suggestions for Authors

This study investigates transcriptomic differences in hippocampal tissue from temporal lobe epilepsy (TLE) patients undergoing anterior temporal lobectomy with amygdalohippocampectomy (ATL/AH). By comparing seizure-free (SF) and non-seizure-free (NSF) cohorts, the research identifies key molecular pathways associated with seizure recurrence, emphasizing immune activation and extracellular matrix (ECM) remodeling.

The study is well-structured, with a robust methodology and a clear presentation of findings. However, several areas need improvement to enhance clarity, scientific rigor, and impact.

My comments: 

  • The study is based on only eight patients, which limits statistical power and generalizability. A larger cohort should be analyzed in future studies to validate findings. Consider discussing how sample size limitations affect the interpretation of results.
  • The study suggests that increased immune activation in SF patients is beneficial, while ECM remodeling in NSF patients is detrimental. However, correlation does not imply causation. Additional functional studies (e.g., in vitro or animal models) should be suggested.
  • The study identifies immunoglobulin genes as potential biomarkers for seizure recurrence. Suggest a validation step using pre-surgical blood tests to assess whether these markers can be used before surgery.

Reviewer 2 Report

Comments and Suggestions for Authors

This paper afford a very challenging topic in epilepsy surgery: the analisys of the postsurgical results in SF vs NSF patients. 'to what extent can differences in gene expression in epileptic hippocampal tissue predict outcome after resective epilepsy surgery?'

Observations:   In clinical practice and in the literature the minimum follow up after epilepsy surgery should be at least of 2 years.

The etiology for the epileptic surgical patients is often unknown. In this series for the patients 7 is 'drug overdose'; this peculiar etiology needs some explanation.

Pages 9 and 10 ,this paragraph 3.7 is related to fig 3 or to fig 6 or to both? Please check.

The paper is very interesting and these preliminary data let us glimpse the possibility to understand the reason why, in apparently similar clinical situations, the surgical results can be both positive or negative. Furthermore the suggestion for pre and post-surgery surveillance could help also in modifying the negative results

Round 2

Reviewer 1 Report

Comments and Suggestions for Authors

The authors revised the manuscript according to my comments.

Thank you. 

I have no comments.

Reviewer 2 Report

Comments and Suggestions for Authors

See previous comments. The authors have addressed all my comments.

Comments on the Quality of English Language

I do not feel qualified to comment on the quality of english